# Antimicrobial and Amyloidogenic Activity of Peptides. Can Antimicrobial Peptides Be Used against SARS-CoV-2?

**DOI:** 10.3390/ijms21249552

**Published:** 2020-12-15

**Authors:** Stanislav R. Kurpe, Sergei Yu. Grishin, Alexey K. Surin, Alexander V. Panfilov, Mikhail V. Slizen, Saikat D. Chowdhury, Oxana V. Galzitskaya

**Affiliations:** 1Institute of Protein Research, Russian Academy of Sciences, 142290 Pushchino, Russia; st.kurpe@gmail.com (S.R.K.); syugrishin@gmail.com (S.Y.G.); alan@vega.protres.ru (A.K.S.); panfilov.alexander@mail.ru (A.V.P.); mikha.shtol@gmail.com (M.V.S.); 2The Branch of the Institute of Bioorganic Chemistry, Russian Academy of Sciences, 142290 Pushchino, Russia; 3State Research Center for Applied Microbiology and Biotechnology, 142279 Obolensk, Russia; 4Department of Biological Sciences, Indian Institute of Science Education and Research Kolkata, Mohanpur 741246, West Bengal, India; saikatduttachowdhury@gmail.com; 5Institute of Theoretical and Experimental Biophysics, Russian Academy of Sciences, 142290 Pushchino, Russia

**Keywords:** amyloidogenic regions, toxicity, antibacterial peptides, proteome, mass spectrometry, severe acute respiratory syndrome coronavirus 2 (SARS-CoV-2), coronavirus disease 2019 (COVID-19)

## Abstract

At present, much attention is paid to the use of antimicrobial peptides (AMPs) of natural and artificial origin to combat pathogens. AMPs have several points that determine their biological activity. We analyzed the structural properties of AMPs, as well as described their mechanism of action and impact on pathogenic bacteria and viruses. Recently published data on the development of new AMP drugs based on a combination of molecular design and genetic engineering approaches are presented. In this article, we have focused on information on the amyloidogenic properties of AMP. This review examines AMP development strategies from the perspective of the current high prevalence of antibiotic-resistant bacteria, and the potential prospects and challenges of using AMPs against infection caused by severe acute respiratory syndrome coronavirus 2 (SARS-CoV-2).

## 1. Introduction

Currently, the relevance of the study and development of antimicrobial molecules is due to the global spread of antibiotic-resistant forms of bacteria, as well as new viral diseases [1,2]. Peptides that exhibit antimicrobial activity have several advantages over conventional drugs, including a broad spectrum of action, slow onset of resistance, and the ability to modulate the host immune response [3]. Several thousand antimicrobial peptides (AMPs) have been discovered, but only a few have been approved by the U.S. Food and Drug Administration (FDA) [4]. Our understanding of the physicochemical properties, the peculiarities of the mechanisms of action of AMPs, modern possibilities of searching and predicting the structure and properties of peptides will define new horizons in the development of new AMPs. Of particular interest is the potential for the use of peptides, which, in addition to pronounced antimicrobial properties, also tend to self-assembly, the formation of supramolecular complexes, and amyloidogenesis [5,6,7]. Traditionally, the amyloidogenic properties of peptides are negatively regarded as contributing to the development of neurodegenerative and progressive metabolic diseases [8,9,10]. However, in recent years, more and more attention has been paid to the study of the functional role of amyloidogenic peptides in the protection of the host and the prospects for their use as antimicrobial agents [11,12]. New data indicate that amyloidogenic peptides can interact with various combinations of pattern recognition receptors (PRRs), which in turn leads to modulation of innate immune cells [13]. Interestingly, such immunomodulatory amyloidogenic peptides can be specific targets for changing entire metabolic pathways. For example, Smole et al. showed that changes in the oligomeric organization of serum amyloid A1 (SAA1) caused by dust mite allergens is recognized by the PRRs and ultimately lead to the development of an immune response [14]. The therapeutic potential of peptide molecules prone to aggregation is truly enormous due to the diversity of their structure and properties, in particular, the ability to form supramolecular complexes with other peptides and proteins, biocompatibility, and biodegradability. It was shown using theoretical and experimental methods that peptide coaggregation is necessary for the destruction of bacterial membranes [15]. Theoretical methods for finding and predicting new AMPs, based on the use of specially designed programs for these purposes, are making an increasing contribution to the development of new AMPs. In turn, we proposed another mechanism of the antimicrobial action of peptides, based on the directed coaggregation of the amyloidogenic peptide with the target protein and the subsequent dysfunction of this protein. Using programs predicting amyloidogenic regions in a protein molecule, sequences of peptides with a tendency to aggregate with the target protein were selected [16]. It is important that the potential for use of peptides combining antimicrobial and amyloidogenic properties is not limited to their use against antibiotic-resistant bacteria. Peptides that tend to coaggregate with viral proteins or prevent the virus from interacting with cell receptors can be used against the spread of coronavirus infection caused by SARS-CoV-2. Currently, significant resources of the scientific community are involved in the search for effective methods of treating coronavirus infection. Many strategies have been proposed, among which peptides play an important role [17]. Interestingly, a similar target effect is a characteristic of AMPs with lectin-like functions (e.g., defensins [18] and hevein-like peptides [19], which can bind to N-glycosylated sites of the viral spike protein or the receptor, which ultimately prevents penetration of the virus into the host cell [20,21]. In general, as will be described below, AMPs have great potential for effective use in the therapy of coronavirus infection [22].

The purpose of this review is to show the possibilities of using the antimicrobial and amyloidogenic properties of peptides to create new therapeutic strategies against antibiotic-resistant bacteria and the spread of viral infections. In the following sections, we describe the physicochemical properties and variety of natural and artificial peptides. We also discuss the known mechanisms of action of antibacterial peptides and our proposed new mechanism associated with directed coaggregation with the target protein of the pathogenic organism. Finally, we present promising results for the search and development of new AMPs, including those against coronavirus infection.

## 2. Brief History of Research, Physicochemical Properties, Classification and Diversity of Antimicrobial Peptides

Speaking about antimicrobial activity, one cannot fail to note the 1945 Nobel Prize winners in physiology and medicine A. Fleming, E.B. Cheyne, and H. W. Florey, who discovered penicillin and its properties. In addition to the discovery of penicillin, A. Fleming described a protein with bacteriolytic activity—lysozyme.

These and other discoveries preceded the “Golden Age of Medicine”, when millions of lives were saved. Antimicrobial substances play an important role in the functioning of the innate immunity of both humans and living beings [23,24]. A significant number of studies on the mechanisms of immune defense have shown the key role of AMPs in the non-specific defense of the host [25]. AMPs are characteristic of all living things: vertebrates and invertebrates, plants, fungi and bacteria [24,25]. Interest in AMPs is mainly due to the fact that most of the known and available methods of treatment were ineffective due to the rapid emergence and spread of resistance in nosocomial bacterial pathogens, fungi and viruses [26]. At the same time, AMPs or host defense peptides have a potent broad-spectrum antimicrobial action against pathogenic bacteria, fungi, and viruses [27]. Thus, AMPs and their various combinations are considered as promising candidates for drugs against multidrug-resistant bacteria such as *P. aeruginosa*, *S. aureus*, *K. pneumoniae* or *A. baumannii* [28,29,30].

AMPs are a group of heterogeneous peptides that have been studied for several decades. During this time, researchers concentrated in several areas of research (Table 1). AMP study is based on the discovery of the antimicrobial activity of peptides in fluids and tissues of humans and mammals [31]. Subsequent studies showed that AMPs play an important role in the immune system of all living beings [32]. The development of physicochemical methods for studying molecular processes made it possible to describe the structural features of peptides and to reveal some connections between the properties of AMPs. The next step, which became a consequence of the computer revolution, was the development of algorithms for predicting the antimicrobial properties of peptides. A century and a half after the discovery of AMPs, out of more than 3000 characterized peptides, seven (oritavancin, gramicidin, daptomycin, colistin, vancomycin, dalbavancin, telavancin) are applicable to clinical practice [4]. Recently, the problems of using AMPs in veterinary medicine and medicine have been widely discussed, which stimulates studies of targeted delivery of AMPs (for example, in the form of metal nanoparticles) or the development of species-specific AMPs [33,34,35]. Undoubtedly, further research will continue the existing directions. However, the topic of AMPs is much broader due to the fact that AMPs are capable of exhibiting antitumor, antibiofilm, antiviral, and other types of activity, the number of studies in these areas is increasing [36,37]. The accumulation of scientific experience and a large number of facts in the study of AMP allows one to focus not only on fundamental issues, but on the possible application of AMP in a practical field. Thus, the described areas are an important basis for further main and secondary research in this area.

In total, 98% of AMPs are encoded in the genome. These peptides can be constitutively produced or induced to maintain the health of the host [24,33]. AMPs are often modified. A total of 24 types of chemical modifications of AMP are known [63]. Three types of modifications are of greatest interest to researchers of drag design: lipidation, glycosylation, and PEGylation. A feature of these modifications is that they increase the activity and bioavailability, increase the metabolic stability of the molecule, and can promote direct translocation [64,65,66]. Lipidation and glycosylation of molecules can be a modification of both natural and synthetic AMPs. Aliphatic chain in peptides modulates their hydrophobicity, tendency to self-assembly, while the possibility of action through an intracellular receptor is not excluded [67,68,69,70]. Glycated peptides also demonstrate greater biostability, the ability to self-assemble complex frameworks, and increase selectivity [71,72]. AMP PEGylation modulates the rate of biotransformation and also reduces toxicity [73,74]. Modifications serve some important functions. For example, enterotoxin AS-48 undergoes cyclization to maintain the correct molecular structure [75]. The cyclical structure of calata B1 supports the functionality [76]. Halogenation and hydroxylation of the arginine, lysine, and tyrosine residues of styelin D allows one to regulate the range of conditions under which this peptide acts [77].

It should be borne in mind that the modification of AMP affects their effectiveness against specific bacteria in different ways. Thus, an analog of AMP with an alternating cationic/hydrophobic structure was optimized, demonstrating only moderate activity against Gram-positive pathogens in [78]. The attachment of hydrophobic fragments to the N-terminus enhanced the antibacterial effect of the analogs against multidrug-resistant *Staphylococcus aureus* and *Enterococcus faecium*. At the same time, modification with hydrophobic fragments led to an increase in activity against Gram-negative *Acinetobacter baumannii*, while the effectiveness against *Escherichia coli* (*E. coli*) strains did not change.

Peptides of different lengths have antimicrobial properties. According to dbAMP data, the polypeptide chain can reach 700 amino acid residues [79]. Interestingly, some peptides with a minimum number of amino acid residues in the chain (5 or less) contain a lipid aliphatic chain [80]. However, 90% of polypeptide chains consist of less than 50 amino acid residues, and most of them contain 20–30 [81]. Considering the hydrophobicity of the molecule, AMPs may consist only of hydrophobic amino acids or not at all. The ratio of hydrophobic amino acids to the total number of residues in the chain may indicate the mechanism of AMP action. Typically, 78% of AMPs have a hydrophobic content in the range of 30–60%. Under physiological conditions, a peptide molecule can be charged depending on its composition. Most AMPs are positively charged—87%, 7% neutral, and 6% negatively charged. Among positively charged peptides, 73% are in the range from +1 to +6, the most pronounced peak corresponds to charge +3. However, the Oncorhyncin II peptide is known to have a total charge of +30. This peptide, isolated from rainbow trout (*Oncorhynchus mykiss*), is the C-terminal fragment of histone H1, which causes pronounced destabilization of flat lipid bilayers [82]. The combination of positive charge and hydrophobicity leads to the formation of an amphipathic nature (spatial separation of the hydrophobic and charged parts of the molecule) of most AMPs. The positive charge is important for the initial interaction with the negatively charged surface of bacteria membranes. The hydrophobic component of the peptide is required for the subsequent attachment of the peptide to the membrane surface.

Like proteins, AMPs form various types of secondary structures (Figure 1): α-helical; containing β-strands stabilized by disulfide bonds; including both α-helical and β-strands, cyclic or loop- forming; linear peptides without definite conformation enriched in proline, tryptophan, histidine, arginine, glycine, or their combinations.

Interestingly, alpha-helical AMPs with a significant frequency contain a small and flexible glycine residue at positions 7 or 14. The results of the study show that although the hydrophobicity of residue 7 and flexibility of the chain at this position can be modulated to increase selectivity, position 14 is less resistant to substitutions [83]. In the extracellular fluids of insects and frogs, α-helical AMPs are present in excess in the form of an extended or unstructured polypeptide chain conformation [84]. These peptides, interacting with membranes, often form a helical fold. The β-region forming peptides have a wide variety of primary structures. Despite the differences, β-fold AMPs share common features, including an amphipathic composition with well-defined hydrophilic and hydrophobic surfaces. Little is known about the structures of proline, arginine and tryptophan rich peptides. There are several conformations that differ from the above. For example, the proline and arginine-rich AMPs and the tryptophan-rich indolicidines form a type II polyproline helix [85,86].

The size and conformation of peptides affect their interaction with membranes [87,88]. In this regard, much attention is paid to the study of the spatial structure and key motifs of folding of AMPs. Recently, in addition to measuring circular dichroism, a high-resolution NMR method has also been used for these purposes [89]. The secondary structure of peptides can be influenced by addition a fluorescent label, which is usually taken into account when developing AMP analogs [90].

AMPs are characterized by the combination of several types of secondary structures in one molecule. The presence of more than one type of structure in the molecule is observed in 94% of known AMPs and, on the contrary, only 4% of peptides have either only helical or β-structural folds [91].

Summarizing the properties and structural features, AMPs are molecules of a peptide nature with a high content of positively charged and hydrophobic amino acids, with spatial separation of charged and hydrophobic parts of the molecule, synthesized in vivo on ribosomes or enzymatically or in vitro, possessing wide activity against bacteria, fungi, viruses, and protozoa in physiological conditions [23,92].

According to G. Wang [81], the world community recognizes two AMP nomenclatures: the first describes the source of isolation (domain, kingdom, and other systematic categories), the mechanism of biosynthesis and properties; the second is based on biological activity.

For classification by source of isolation, the biological nomenclature proposed by Robert H. Whittaker in 1969 is used. According to this nomenclature, the detected AMPs are distributed as follows: prokaryotes (bacteria and archaea) include 272 and 4 peptides, respectively, protists (protozoa and algae)—8, fungi—13, plants—335, and animals—2043 [81]. As of March 2020, over 70% of the peptides in the CAMPR3 electronic antimicrobial peptide database come from the animal kingdom. It is Interesting that in this kingdom two main sources of AMPs can be distinguished: 50% of all sources are amphibians, 13% are insects [63].

Bacterial peptides are grouped into the bacteriocin class. Among the known AMPs, there are about 6% of them, and the number of bacteriocins is growing rapidly, since new representatives of this class are being developed [93]. The main systematic feature of the classification of bacterial AMPs is the structure of the cell wall of bacteria producing peptides. The subsequent classification is carried out according to the features of the structure of the peptide and its molecular weight. There are three classes of peptides synthesized by Gram-positive bacteria. The first class—lantibiotics—is characterized by the presence of thioether rings in the molecule. There is no such structure in second-class AMP molecules. These peptides are classified into 4 subclasses: pedoicin-like bacteriocins (subclass IIa), two-component bacteriocins (subclass IIb), circular bacteriocins (subclass IIc), and linear unmodified pedicin-like bacteriocins (subclass IId). Gram-negative bacteria are classified in a similar way [81]. Bacteriocin of Gram-negative bacteria is grouped into four classes [94]. Colicins and colicin-like peptides have similar physicochemical properties and differ only in the type of producer bacteria. These AMPs have a high molecular weight and are sensitive to temperature and proteases. The synthesis of colicins is associated with the expression of cell autolysis proteins. The mechanism of action can be directed both to the cell membrane and to intracellular targets. Tailocins are multicomponent peptides with a molecular weight of 20 to 100 kDa, consisting of 8-14 different polypeptide chains homologous to the modules of the bacteriophage tail. The exact mechanism of action of the tailocins is not known, but is directed against a narrow group of related strains. Microcins are low molecular weight peptides (<10 kDa) that are resistant to proteases, extreme pH and temperature, and can have complex post-translational modifications. The mechanism of action can be explained both by the formation of pores in the membrane and by inhibition of DNA gyrase and RNA polymerase [94].

The AMPs of the fungal kingdom are divided into two classes. Peptides isolated from soil fungi of the genera Trichoderma and Emericellopsis form the peptaibole class. These molecules are characterized by the presence in the composition of 15–20 amino acids with a high content of aminoisobutyric acid, and acetic acid residue at the N-terminus and hydroxyamino acid at the C-terminus. The second class is represented by defensin-like peptides, usually containing multiple disulfide bonds.

Antimicrobial plant peptides are classified into 9 groups [81]. The key features of these AMPs are high content of cysteine and glycine in the molecule, as well as the presence of disulfide bridges to increase the stability of the structure in stressful conditions. About 17% of amino acids in plant AMPs are charged mainly due to arginine and lysine.

However, it is interesting to note that residues such as aspartic acid and glutamic acid may also be present. The presence of negatively charged amino acids can play a significant role in the activity against pathogenic bacteria [95].

The classification of animal AMPs is complex, and the number of systematic units is extensive [81]. Amphibians have the following AMP families: magainins, dermaseptins, brevinins, esculentins, japonitsins, nigrocin-2, palustrins, ranatsicins, ranatuerins, and temporins [96,97]. In insects, the cetsropin, defensin, and proline-rich peptide families are well known [98]. With regard to marine invertebrates, Otero-González et al. [99] described AMPs of various types, such as Porifera, Cnidaria, Mollusca, Annelida, Arthropoda, Echinodermata, and Chordata. In mammals, including humans, the main AMP families are defensins and catelicidins [100].

The disadvantage of classification by the source of isolation is the impossibility of assessing the structural similarity of molecules of different systematic categories. Therefore, a kind of artificial system is created; on the bases of genomic and proteomic data, AMPs were divided into 45 families. This classification is based on the similarity of the primary structure of the peptide [91].

Defensins hold a special place among the known families. A superfamily of small (4–6 kDa) cationic peptides secreted by many species, including humans and other mammals, as well as fish, birds, insects, fungi, and plants. The persistence of this superfamily of peptides among evolutionarily young and ancient groups indicates that peptides are an ancient and conservative defense mechanism [101]. Defensins are synthesized in the cell as precursors. Interestingly, the precursor plays a key role in the functioning of the peptide. For example, the α-defensin precursor contains 40 amino acids and is required to maintain a biologically active peptide in an inactive state.

By interacting with a mature peptide, the pro-segment acts as an effective intramolecular inhibitor, avoiding autocytotoxicity [101]. Another widely represented AMP family is thaumatins. Peptides of this family exhibit antifungal activity. Thaumatins have been identified in fungi, plants, and animals (nematodes, ticks and insects). Comparison of thaumatins of animal and plant origin reveals their close similarity, which also indicates the antiquity of both the AMP family and the defense system itself [102]. Typical plant thaumatins have a molecular weight of about 20 kDa, an even number of cysteine residues (10–16), a conserved REDDD sequence in the primary structure, and a conservative three-dimensional organization [103].

AMPs provide a wide range of biological functions (see Figure 2A). In addition to antimicrobial, antibacterial, antiviral, antifungal, antiparasitic activity, AMPs may have more specific effects (see Figure 2B).

Antimicrobial activity is understood as the presence of one or more, and in rare cases all types of activity that suppress the life of viruses, bacteria, fungi or parasites [81]. It is known that the amount of AMP decreases with an increase in the area of their activity. A decrease in the number of activities may be associated with the physicochemical properties of peptides [81]. Indeed, there are very few well known AMPs with all four types of activity. These peptides are amphibian magainin 2, dermseptin C1 and dermseptin C4; insect melittin; human α-defensin HNP-1, cathelicidin LL-37; bovine BMAP-27, BMAP-28 and vegetable calata B2 [96,100,104,105,106,107,108]. The antibacterial activity of peptides is the most common. This activity was found in 31% of the peptides in the dbAMP database. This activity is usually explained by the action of cationic peptides on the membrane [81].

Interestingly, the activity against Gram-negative and Gram-positive bacteria is distributed approximately equally (19% and 22%, respectively). 16% of dbAMP peptides have antifungal activity. This activity often belongs to representatives of plant AMPs [63]. The antiviral activity of AMP is interesting. Cathelicidin LL-37 is known to be active against human immunodeficiency virus 1 (HIV-1), respiratory syncytial virus (RSV) and influenza virus [11,109]. Moreover, viral particles, both coated and uncoated, are exposed. A possible mechanism of action is associated with the ability of the antimicrobial peptide LL37 and similar α-helical AMPs to form nanocrystalline complexes with viral dsRNA. Such complexes are recognized by toll-like receptors, which further induce an immune response [110,111].

It is assumed that the amphipathic nature of AMP allows it to bind to the viral nucleic acid. Antiparasitic peptides make up a relatively small group of AMPs. Although parasitic organisms are usually multicellular eukaryotes, the mechanism of action of antiparasitic peptides involves cell death trough destruction of the cell membrane, as in other AMPs [112]. Peptides in this group are active against various parasites, including *Leishmania* and *Trypanosoma brucei* [113].

AMPs may be of particular interest in medicine. The anti-cancer activity of AMP is of special interest. Several studies have shown that some AMPs are cytotoxic to cancer cells [114,115,116]. This discovery is of great importance, since there is a problem of resistance of cancer cells to anticancer drugs [117]. It is not yet clear why some peptides are capable of killing cancer cells while others are not. It is assumed that there is some similarity between cancer and bacterial cells, which allows AMP to specifically bind to them without exhibiting cytotoxicity towards healthy cells [118].

Also, of great interest is the activity of AMPs against microbial biofilms. In addition to drugs, various peptide conjugates can be used to coat catheters to prevent the formation of polymicrobial biofilms [119]. Bacteria in biofilms are coated with a complex mixture of extracellular polymeric substances. In such an environment, bacteria become extremely resistant to common (conventional) antibiotics and the host’s immune response [120]. It has been shown that human cathelicidin LL-37 is able to inhibit the biofilm formation by *Francisella novicida* [121]. Another artificially modified peptide R17, in addition to inhibition, exhibited dispersing activity against biofilms of *Acinetobacter baumannii* [122]. The DP7 peptide was shown to inhibit the formation of multidrug-resistant *Pseudomonas aeruginosa* biofilms [123]. In general, the methods of destruction of biofilms using AMPs are currently being intensively developed [124]. AMPs can be used as a treatment option for patients with recurrent infections who will benefit from high local doses of persistent antimicrobials. An important way to improve AMP is to decrease the value of their minimum inhibitory concentration (MIC) in relation to the pathogen. The approaches to reducing the MIC can be as follows: selection of the cyclization point, modification of amino acid residues taking into account changes in their hydrophobicity, as well as changes in the length of the cationic side chain, and various combinations of these approaches [125]. 

A contribution to understanding the action of AMP can be made by creating analogues of AMP and studying their action. For example, the study of analogs of cathelicidin allowed researchers to reveal patterns in the manifestation of its antimicrobial action, depending on the characteristics of the amino acid sequence. Thus, Ser9 in cathelicidin LL-23 was replaced by Ala or Val, which made LL-23 hydrophobic and enhanced its antibacterial activity. A decrease in the rate of hydrogen-deuterium exchange from LL-23 to LL-23A9 to LL-23V9 suggests a deeper penetration of LL-23V9 into micelles, which correlates with an increased effect [126]. LL-37, which converts positively charged arginine to neutral citrulline, in contract to the unmodified peptide, does not kill *E. coli*, indicating that the net positive charge is important for the antibacterial and membrane lysing effects [127]. It has also been shown that citrullination of LL-37 reduces its direct antiviral activity against human rhinovirus. In addition, it is discussed that overexpression of the gene of peptidyl arginine deiminase (PAD) and citrullination of the host defense peptide (HDP) during infection may represent a new evasion mechanism for the virus [128].

Various strategies have been proposed for surface immobilization of a wide variety of materials and local release of AMPs from implant surfaces [129]. For LL-37 a significant loss of antimicrobial function was shown when the peptide was exposed to carbon nanoparticles at low concentrations, and the interaction of nanomaterials with the peptide led to a significant change in the peptide structure [130]. Another study showed that encapsulation of the AMP in dextran nanoparticles increases the residence time in the lungs of the peptide administered via aerosol [131]. In a recent work, the effective localization of natural AMPs on a negatively charged bacterial surface was described using structurally nanoengineered AMP polymers (SNAPPs) [132]. In addition, it was proposed to use cubosomes for local delivery of AMPs [133]. It has been shown that the creation of conjugates based on antibiotics and AMP is effective against *Staphylococcus aureus* [134]. The green fluorescent protein (GFP) based fusion lantipeptide class I expression system is a robust system with the advantage of direct visualization of expression and purification by GFP fluorescence [135]. Small ubiquitin-related modifier (SUMO) technology is widely used in *E. coli* expression systems for the production of AMPs. However, *E. coli* is a pathogenic bacterium that produces endotoxins and often secretes target proteins to form inclusion bodies. *E. coli* was replaced by the *Bacillus subtilis* expression system using SUMO technology to synthesize cathelicidin-BF (CBF), an AMP purified from venom of *Bungarus fasciatus* [136].

## 3. Mechanisms of Antimicrobial Action of Peptides

### 3.1. Analysis of Possible Mechanisms 

The wide variety of AMPs and their functions indicates the complexity of the defense mechanisms. It is believed that the selectivity of AMPs action is regulated by production inducibility and limited localization at the lesion sites or by the nature of the peptide [84]. The selectivity of AMPs can be determined by the following physicochemical and structural parameters: conformation, charge, hydrophobicity, hydrophobic moment, amphipathicity. It is essential that these molecular parameters are interdependent, therefore, a change in one parameter often leads to compensatory changes in others. Despite the great variety of the primary structure, AMPs form typical elements of the secondary and tertiary structure. These features are important for systematization and classification (see Figure 1). 

The total charge of AMPs plays an important role in the initial electrostatic attraction of molecules to the membranes of bacteria and other microorganisms. It is the negative charge that forms those structural features of the bacterial membranes that provide selectivity in relation to host tissues. The presence of phosphatidylglycerol, phosphatidylserine, cardiolipin provides total negative charge, and the presence of lipopolysaccharides, teichoic and teichuronic acids gives an additional negative charge to the membranes of Gram-negative and Gram-positive bacteria. The chemoosmotic potential of a prokaryotic target cell is usually 50% higher than that of mammalian cells. Some authors emphasize the correlation between the total positive charge of the peptide and its antimicrobial activity [137]. However, there is a limit beyond which an increase in the positive charge does not lead to an increase in the activity of the peptide. Indeed, an increase in the charge of the AMP magainin from 3 to 5 leads to an increase in antibacterial activity against Gram-negative and Gram-positive pathogens; however, increasing the charge to 7 significantly increases the hemolytic activity and leads to the loss of antimicrobial activity [138].

Among the physical characteristics reflecting the hemolytic activity of AMPs are amphipathicity and hydrophobic moment. The hydrophobic moment is a quantitative characteristic of amphipathicity, calculated as the vector sum of the hydrophobicity of individual amino acids normalized to an ideal helix [139]. An increase in the hydrophobic moment leads to a significant increase in the permeability and hemolytic activity of model peptides in relation to target membranes. In a study by Dathe et al. [140], a relatively small increase in the hydrophobic moment led to an 8-fold decrease in the amount of peptide required for hemolysis.

Another important characteristic of the interaction of AMPs with membranes is hydrophobicity. This property determines the ability of AMP to be incorporated into the lipid bilayer. In a study by Dathe et al. [140] designed peptides with constant charge, helicity and hydrophobic moment, but differed in hydrophobicity. With a slight change in the hydrophobicity of the molecules, they achieved differences in membrane binding and permeability. This indicates the importance of hydrophobic properties in the interaction of the peptide and the lipid bilayer. Thus, for peptide S3(B), the effects of changing the overall hydrophobicity of the modified peptide were shown due to the introduction of fatty acid fragments of different lengths and in different positions into this cyclic peptide containing a flexible linker [141].

The main activity of AMP is a membranolytic mechanism aimed at disrupting the integrity of the cell membrane and cell wall of bacteria. At the moment, five such mechanisms are known: the threshold concentration mechanism, the conformational phase transition mechanism, the keg mechanism, the toroidal pore mechanism and the “carpet” mechanism. In this review we suggest a new mechanism based on the direct coaggregation of amyloidogenic peptide with its protein or protein silencing mechanism.

### 3.2. Mechanism of the Threshold Concentration

The mechanism of the threshold concentration includes two stages of the interaction of AMP with the cell. First, the peptide under the action of electrostatic forces binds to phosphate groups in the membrane and accumulates on the surface. After reaching a certain concentration, the second stage begins. Peptides begin to penetrate and pass through the lipid bilayer, binding to intracellular targets. It is assumed that the membrane potential electrophoretically draws cationic peptides into a non-polar membrane environment, thereby reducing the energy barrier for pore formation [142]. In addition, peptides can multimerize and thus form pores. A recent modeling study also showed that the orientation of Mac1 AMP remains transmembrane in bilayers and on micelle surfaces regardless of lipid charge. Thus, as suggested by the authors, the orientation of the peptide apparently, depends more on the curvature of the membrane than on the surface charge [143]. In addition to the concentration of the peptide and the tendency to self-assembly, the phospholipid composition of the membrane and its fluidity can be factors influencing the rate-limiting stage of the process.

### 3.3. Conformational Phase Transition Mechanism

The conformational phase transition mechanism involves a change in the structure of the peptide molecule after binding to the membrane. Numerous studies using various biophysical methods show that disordered AMPs exhibit extended or random helical conformations in aqueous media under conditions of a bilipid layer [144]. Interestingly, the composition of the membrane can determine the possibility and rate of conformational transition of the peptide. For example, the frog skin PGLa peptide requires a negatively charged phosphatidylglycerol and phosphatidylethanolamine bilayer to make this transition. Compared to helical peptides, β-structured peptides containing disulfide bonds are generally more stable in polar and non-polar solvents. However, some quaternary peptides dissociate upon contact with the membrane surface. The selective toxicity and antimicrobial activity in this case can be explained by monomerization of oligomeric peptides.

### 3.4. Keg Mechanism

The keg mechanism implies the action of AMP through the formation of pores in the cell membrane. Pore-forming peptides are located in the membrane in a barrel-like form. The keg walls are formed by individual peptides or peptide complexes. The β-regions and α-helices face outward and together with the acyl chains of the bilipid layer form a hydrophobic surface, while the pore channel is formed by a hydrophilic surface [145]. The initial stage of pore formation involves the binding of peptides, probably in the form of monomers, to the membrane surface (see Figure 3A). Upon binding, the peptide can undergo a conformational phase transition, causing a thinning of the membrane due to the redistribution of the polar heads of phospholipid groups. At this stage, the hydrophobic part of the peptide penetrates the membrane. In addition, the peptide accumulates in the bilipid layer until the concentration reaches the threshold required for aggregation and incorporation into the hydrophobic membrane core. The transmembrane configuration of the peptides provides minimal impact on the membrane surface, and the pore expansion is associated with the attachment of peptide monomers. An example of such a mechanism of action was proposed for alamethicin [146,147]. Alamethicin-induced membrane permeability changes stepwise with continuous increase in peptide concentration. This fact indicates the presence of pores of different diameters corresponding to channels consisting of four or more transmembrane peptides.

### 3.5. Toroidal Pore Mechanism

One of the well-studied mechanisms of peptide-membrane interactions is the toroidal pore mechanism. The main difference between this mechanism and the «barrel-strand» mechanism is the intercalation of peptides by membrane lipids (Figure 3B). In this model, peptides in the extracellular environment acquire a helical structure due to interaction with a charged and hydrophobic bacterial membrane. Initially, the helices are oriented parallel to the membrane surface, which is confirmed by the data of NMR, fluorescence, and circular dichroism [148]. Hydrophobic residues of bound peptides displace the polar heads of phospholipids, creating a rupture in the hydrophobic region and causing membrane deformation [148]. The resulting pressure and thinning further destabilize the membrane surface, making it more vulnerable to subsequent peptide interactions. When the threshold concentration is reached, peptides are oriented perpendicular to the membrane, which was demonstrated for magainin at a peptide/lipid ratio of 1:30 [84]. At a threshold concentration, peptides tend to self-association, which leads to the formation of a peptide/lipid-supramolecular pore complex. Pore dissociation leads to the penetration (translocation) of peptides into the cytoplasm and (their) binding to intracellular targets. The characteristic features of toroid pores are finite channel lifespan, discrete size, ionic selectivity, and an inverse relationship between peptide stability and peptide charge [149,150,151].

### 3.6. The “Carpet” Mechanism

The “carpet” mechanism is characterized by membrane damage without penetration of AMP into the cell (Figure 3C). In this model, the target membrane is covered with an extensive layer of peptides. Changes in the mobility of phospholipids due to molecular architecture and membrane lead to and/or a decrease in the barrier properties of the membranes and subsequently to the destruction of membranes [152]. As in other models, peptides initially bind to the membrane mainly through electrostatic interactions, covering the phospholipid bilayer [153]. Upon reaching the threshold concentration of the peptide, the membrane collapses due to energy losses. Thus, membrane disruption occurs without channeling, and peptides do not penetrate into the hydrophobic membrane core. The AMP cecropin P1, which is characterized by such a mechanism of action, is initially oriented parallel to the membrane and does not penetrate into the hydrophobic medium. This orientation destabilizes the packing of phospholipids and causes membrane destruction when a concentrated layer of peptide monomers is formed on the surface [154]. Likewise, the tryptophan-rich peptide indolicidin does not penetrate the membrane according to the data of fluorescence spectroscopy, but it is an effective AMP [155].

In recent years, many authors have discovered intracellular targets for the action of AMPs. For example, the mechanism of action of human β-defensin 3 is not based on the formation of specific transmembrane pores, but on interference with the organization of membrane-bound multienzyme mechanisms, such as the cell wall biosynthesis complex and the electron transport chain [156,157]. Fungal defensin eurocin inhibits cell wall synthesis by binding equimolar to the cell wall precursor lipid II [158]. Comparison of the mechanisms of action of the two lipopeptides showed that the lipopeptide compound INA-Ac-5812 directly destroys the integrity of the membrane, while daptomycin exhibits an antimicrobial effect, binding to methicillin-resistant Staphylococcus aureus cells and interrupting cell wall biosynthesis with followed by delocalization of peptidoglycan components of the biosynthetic apparatus [159,160]. It has been shown that AMPs are capable of altering DNA morphology, with possible effects of nucleic acid disruption and fragmentation [161]. For example, buforin II and takiplesin bind to DNA, pleurocidin and indolicidin inhibit the DNA, RNA and protein synthesis (biosynthetic processes), drosocin and pyrrhocoricin reduce enzymatic activity, mersacidin inhibits the biosynthesis of the cell wall [162]. The study of supramolecular structures formed between DNA and AMPs has demonstrated a correlation between these structures and their ability to activate toll-like receptor 9 (TLR9), an important receptor expressed in cells of the immune system [163]. As mentioned above, AMPs exhibit a wide range of biological activities (Figure 2). Moreover, a number of AMPs are known to perform two or more inhibitory functions, which may indicate the action of one peptide against different targets [164]. Therefore, it is important to consider both membranolytic and intracellular mechanisms of AMP action.

It was noted that the action of some AMPs does not lead to cell lysis. One of these peptides is indolicidine. This peptide inhibits only DNA biosynthesis. The targets of indolicidine are regions of single- or double-stranded DNA with deleted nitrogenous bases. The peptide was found to have a high affinity for double-stranded (ds) [AT], ds [CG] and ds [AG], but less affinity for ds [GT]. The binding of indolicidine to duplex DNA is stabilized by the central Pro-Trp-Trp-Pro motif, thus DNA replication and transcription are inhibited. The ability of indolicidine to inhibit DNA topoisomerase was also noted [165]. Another peptide, microcin 25, synthesized by E. coli, is able to bind to the catalytic site of RNA polymerase, which temporarily stops the elongation of the transcript. It is assumed that microcin 25 can bind to the 3′-end of the newly synthesized template and block the elongation complex of GreA/GreB-dependent transcription [166].

AMPs can also hamper processes of protein synthesis. The translation system can be blocked not only by inhibition of chaperones, translation factors or ribosomes, but also at the transcriptional level. The interaction of the peptide with any of the associated enzymes or effector molecules can interrupt protein biosynthesis. It is reported that Bac7 peptide, isolated from neutrophil granules, in addition to its membranolytic activity, inhibits translation when interacting with ribosomes [167]. Proteome analysis revealed that E. coli treated with Bac7-demonstrated enrichment in a number of metabolic pathways, including nucleic acid metabolism (transport and metabolism of nucleotides) [168].

Another interesting mechanism of the antimicrobial action of peptides is the inhibition of the protein folding system. DnaK is the main bacterial heat shock protein 70 (Hsp70), which plays a key role in the folding process in E. coli cells. It should be also noted that, in addition to inhibiting protein biosynthesis, Bac7 has a high binding constant with DnaK and inhibits the protein folding function in the DnaK-DnaJ-GrpE-ATP molecular chaperone system [169]. Proline-rich AMPs (pyrocorticin isolated from the bug *Pyrrhocoris apterus* and apidaecin from the beetle *Riptortus clavatus*) have the DnaK-binding YL/IPRP motif. Both pyrocorticin and apidecin inhibit DnaK and GroEL, and also disrupt the ATPase activity of DnaK [170,171]. Another proline-rich AMP, oncocin, consisting of 19 amino acids and obtained from the antibacterial peptide oncopeltus, exhibits high antibacterial activity against Gram-negative bacteria. The peptide easily penetrates the bacterial membrane without any lytic effect on cell. Oncocin has been shown to form a complex with DnaK, which indicates an inhibitory activity against protein folding [172].

### 3.7. Mechanism of Direct Coaggregation or Protein Silencing

Here we should talk about the mechanisms of the antimicrobial action of amyloidogenic peptides [12]. Most studies of the antimicrobial mechanisms of action of amyloids reveal their destructive effect on cell membranes, for example, through channel/pores formation or membrane thinning [12,173,174,175]. Several studies have demonstrated the formation of antibacterial nanonets by amyloids, which inhibit the spread of microbes [176,177]. AMPs capable of forming hydrogels with mechanical protective properties for the further proliferation of both planktonic bacterial cells and biofilms are of particular interest [178,179]. Using the software AGGRESCAN [180], FoldAmyloid [181], PASTA 2.0 [182] and Waltz [183], we searched for and predicted amyloidogenic regions in a sample of 300 and 499 AMPs showing activity against Gram-negative and Gram-positive bacteria, respectively (Figure 4).

Figure 4 demonstrates similar patterns of distribution of predicted amyloidogenic regions in AMPs, showing activity against Gram-negative and Gram-positive bacteria, respectively. This may indicate that the ability to aggregate may be combined with antimicrobial action against Gram-negative and Gram-positive bacteria. Interestingly, amyloidogenic regions are predicted in about half of the AMP samples. The median length of the AMP was 30 amino acid residues (a.a.), and the median length of the amyloidogenic region was 17 a.a. residues. At the moment, there is no theory about how antimicrobial activity and aggregation ability are related.

At the same time, we would like to consider a new mechanism of the antimicrobial action of amyloids based on the targeted blocking of the functions of a key protein, the so-called protein silencing. This mechanism is based on the phenomenon of coaggregation of proteins and peptides [184,185,186]. Thus, it is known that human serum albumin is able to form coaggregates with the amyloid β (Aβ) peptide due to binding sites that are evenly distributed over three domains on the protein surface [187,188]. Human serum albumin, like many other multi-domain proteins (immunoglobulin light chains [189,190], titin [191,192], YB1 [193,194], tends to aggregation and is capable of forming fibrils [195].

Moreover, the process of aggregation/fibrillation of human serum albumin involves changes in the tertiary structure of domains II and III [196]. It was shown that human serum albumin itself can prevent the formation of amyloids not only for the Aβ peptide, but also for insulin [197], and the regions of this protein that are responsible for interaction with amyloids were identified [198,199]. Studies of the formation of amyloids and aggregation sites made it possible to hypothesize site-specific coaggregation of the target protein with an amyloidogenic peptide as one of the possible antibacterial mechanisms of AMP action. 

The largest multi-domain and functionally important bacterial ribosomal S1 protein was proposed as a target protein, the disruption of which is critical for the life of bacteria [200,201,202]. The full-length S1 protein has a high mobility of individual protein domains and a tendency to aggregation, which creates problems for establishing its three-dimensional structure [203,204]. At the same time, using computational and experimental approaches, we described specific amyloidogenic regions within the domains of the S1 protein, which can be potential sites for modulating the aggregation properties of bacterial ribosomal S1 proteins [9,205,206,207]. Based on the predicted amyloidogenic regions, amyloidogenic peptides were synthesized, some of which exhibited antimicrobial activity in vitro tests [16].

The mechanism proposed by us can be divided into several stages. The first stage is the penetration of the effector molecule into the cytoplasm of the bacterial cell. We assume that the penetration of peptides occurs by analogy with peptides penetrating into cell [208]. Penetration through the bilipid layer can be carried out in several ways. One of the methods involves direct translocation of the peptide across the membrane and its local disturbance without further destruction. [209]. The peptide crosses the bilipid layer and enters the cytoplasm due to the transmembrane potential, pH gradient, and other physicochemical processes [163,210]. Another way of peptide penetration through the membrane can be via by endocytosis and/or macropinocytosis [211,212].

The second stage is the binding of the effector molecule and the target protein. The action of the effector molecule can lead to both bacteriostatic and bactericidal action, depending on the chosen target. The target protein must perform several vital functions in the cell, which are disrupted when this protein coaggregates with an effector peptide. To assess possible binding, various approaches can be used both in silico based on interaction patterns [213,214], and in vitro using spectrophotometry, NMR, or electron microscopy [215,216,217]. 

The third stage is the aggregation of the effector molecule and the target protein. Self-aggregation of a protein or its coaggregation with an effector peptide should lead to blocking of protein functions and a decrease in the viability of the bacterial cell. In our study, we used methods of thioflavin T fluorescence, electron microscopy, spectrophotometry and mass-spectrometry to assess the amyloidogenic and antibacterial activity of peptides synthesized on the basis of the ribosomal S1 protein from *Thermus thermophilus* [16]. It has been experimentally demonstrated that amyloidogenic regions of the ribosomal S1 protein can be used as candidates for the development of new antibacterial peptides [16].

Coaggregation of the target and effector can occur by the mechanism of aggregation controlled by nucleation, which has already been demonstrated for the Aβ(1-40) peptide during the formation of amyloid fibrils [218]. Interestingly, Aβ peptides isolated from the brains of Alzheimer’s disease patients have amyloidogenic properties as well as antimicrobial activity, and that incubation of these brain samples with antibodies to Aβ reduced the antimicrobial activity of the peptides. Several studies report that Aβ peptides have demonstrated antibacterial and antifungal activity, as well as antiviral effects, in particular against influenza virus. At the same time, as the authors note, Aβ(1-42) has a greater antimicrobial or antiviral activity than Aβ(1-40), which suggests a possible relationship between the ability of Aβ(1-42) to assemble into oligomers and its antiviral activity, since this peptide has a greater tendency to form oligomers and fibrils than Aβ(1-40) [11]. 

## 4. Prediction and Development of New AMPs

### 4.1. Types of Programm for Prediction and Development of New AMPs

Over the past decades, great efforts have been made to discover AMPs in natural sources using wet biology techniques. Classical chromatographic methods for the detection of natural AMPs have played an important role in shaping the modern view of AMP in terms of biological source, amino acid sequence, three-dimensional structure, and antimicrobial activity. Since the 90s of the twentieth century, there are more and more possibilities for predicting and designing AMP with a specific function, pharmacological target and activity. The first report of such a discovery was associated with human cathelicidin, this peptide was detected using the method of the conservative sequence alignment of the pro-peptide domain [219]. In 2004, Yount and Yeaman discovered a motive for identifying new AMPs [220]. In addition, more efficient genomic and proteomic methods for the identification of peptides with antimicrobial function have been developed for peptide in recent years [221,222]. The application of prediction methods to genomes and proteomes, followed by experimental verification, accelerates the rate of discovery of peptides, which, in turn, allows expanding understanding of the functional nature of these compounds.

The similarity of peptides with and without antimicrobial activity is a bottleneck for bioinformatics methods. To solve this problem, programs for predicting antimicrobial activity based on various mathematic algorithms and information sources are used.

The programs of the first category use only data on the primary structure of mature AMPs. Based on these data, the G-X-C motif was discovered in peptides containing disulfide bonds (defensins) [223]. Using this motif, Yount and Yeaman discovered previously unknown peptides brazzein and charybdotoxin, which are active against bacteria and fungi *Candida albicans* [220]. A total of 28 new human defensin was identified. Programs in this category determine the length of the peptide, its charge, the content of hydrophobic residues in the molecule, and its amino acid composition. The predictions are made according to the correspondence of these peptide parameters to the range of parameters from the database. Machine learning methods are also used for prediction. Artificial neural networks perform highly accurate predictions based on rules from databases of antimicrobial peptide.

The programs of the second category use highly conserved sequences of amino acid residues of pro-peptides. The amino acid sequences of mature peptides are more variable than the sequences of pro-peptides. This observation lies in the strategy of searching for such peptides [100]. Since the discovery of the first (antimicrobial) peptide, cathelicidin, in 1988, more than 100 such peptides have been identified. Analysis of processing signals allows predicting antimicrobial activity of other classes of AMPs. For example, amphibian antimicrobial peptide precursors also have a common and highly conserved pro-region, usually terminating in a typical Lys-Arg processing signal. This discovery was used as the basis for identifying many amphibian peptides [222,224].

A combination of approaches from previous types of programs is used in the third category. Fjell et al. created a program for prediction based on both pro-peptide sequences and mature peptides [225]. Peptides from Antimicrobial Sequences Database (AMSDb) are classified into several clusters, after which prediction parameters are calculated [226]. The authors identified 146 clusters for mature peptides and 40 clusters for pro-peptides. Using this program, it was possible to achieve 99% prediction accuracy. However, it is unclear how many more clusters can be found based on the most recent collection of naturally occurring AMPs [63], and whether it is possible to make the prediction even better.

The fourth type of programs is based on the search for homologous sequences of enzymes that carry out the transfer or modification of AMPs. LanM is a group of proteins that modify peptides from the lantibiotics family. The search for homologs of this group of proteins led to the discovery of new peptides in this family—haloduracin and lichenicidin [227]. Using this approach, 89 LanM homologues were identified; some of them were found in 61 strains that do not produce lantibiotics. The conserved LanT transporter was used to detect other lantibiotics. Based on the similarity with this protein, Singh and Sareen identified 54 bacterial strains containing LanT homologues [228]. Morton et al. used a protein cluster required for modification, transport, and resistance to bacteriocins for prediction [229]. Using this program, the authors predicted bacteriocin gene blocks for 2773 genomes.

The last type of program uses genetic information about expression, processing, and transport, and also performs comparisons with the already described AMPs. The processing of eukaryotic transcription data using data mining tools has allowed the identification of new AMPs. Lynn et al. identified 9 novel chicken AMPs by searching for homology tags of clustered expressed sequences using BLAST and a hidden Markov model [230]. Amaral et al. identified about a hundred AMPs based on physicochemical characteristics: peptide length, total charge surface, and hydrophobic moment [231]. Proteolytic cleavage of proteins can release AMP. Based on this fact, Torrent et al. developed a theoretical method for identifying potentially active regions with antimicrobial activity [232]. Hellinger et al. achieved significant success by combining transcriptomic and proteomic data [233]. They were able to identify 164 AMPs from single plant: 108 based on transcriptomic data only and 127 based on mass spectrometry data. Perhaps one of the most effective strategies for searching of natural candidates for AMPs is based on in silico methods of analysis of RNAs synthesized after immunization of host cells with bacteria [234].

It is important to note that in order to predict AMPs that will act by the mechanism of coaggregation with the target protein (Figure 5), it is first of all necessary to search for amyloidogenic regions in this protein [205,235]. Approaches that include algorithms to predict aggregation/amyloidogenic sites may be useful for this purpose [180,181,182,183]. In addition, information on specific folded/unfolded or rigidity/flexibility regions of the protein may be appropriate for the development of AMP based on the sequence of the target protein [236,237].

It is important to take into account AMP development programs that are used to predict a possible immune response and allergic reactions to a specific peptide structure [238]. This validation is especially important for drug development based on AMPs, in particular, antiviral drugs, which will be discussed in more detail in the final section of this review.

### 4.2. SARS-CoV-2 Like an Object for Prediction and Development of New AMPs

Coronaviruses (CoVs) are a diverse group of viruses that infect a variety of animals, including live animals, including livestock, poultry, and can cause mild to severe respiratory infections in humans. Severe Acute Respiratory Syndrome Coronavirus 2 (SARS-CoV-2) is a novel highly transmissible, pathogenic coronavirus that emerged at the end of 2019 and triggered a pandemic of new acute respiratory disease, also known as coronavirus disease 2019 (COVID-19). Previously, Severe Acute Respiratory Syndrome Coronavirus 1 (SARS-CoV-1) and Middle East Respiratory Syndrome Coronavirus (MERS-CoV) caused an outbreak of unusual viral pneumonia in 2002 and 2012, respectively [239]. 216 countries and regions from all six continents have reported more than 65 million COVID-19 cases, and more than 1.5 million patients have died, according to data from www.worldometer.info on December 4, 2020. The spread of SARS-CoV-2 is currently a serious threat to the health and life of people around the world. The efforts of scientists and specialists are aimed at developing effective methods of treating coronavirus infection. Clinical trials of various treatments for SARS-CoV-2 are currently underway, but none have been approved yet. Thus, SARS-CoV-2 is a relevant object for demonstrating the features of a strategy for the development of AMPs against a specific virus. It is well known that some peptides have antimicrobial activity against viruses. For example, melittin has significant antiviral activity and reduces the infectivity of enterovirus, human immunodeficiency virus (HIV), influenza A viruses, vesicular stomatitis virus (VSV), and some other viruses [240]. Thus, AMPs, derived from cathelicidin, effectively suppress Ebola virus the infection [241].

The genome of SARS-CoV-2 is around 30 kb, and is a positive-sense single-stranded RNA (+ssRNA) that contains six functional open reading frames (ORFs): spike (S), nucleocapsid (N), membrane (M), envelope (E) and replicase (ORF1a/ORF1b) [242]. The SARS-CoV-2 S protein consists of two subunits. The S1 subunit is important for binding to the host cell angiotensin-converting enzyme 2 (ACE2) receptor and the S2 subunit is responsible for the fusion of the virus and the cell membrane [243]. The S1 region is further divided into two functional domains, N-terminal domain (NTD) and three C-terminal domains (CTDs) [244]. CTDs contain a region (amino acids 319–529) of the receptor-binding domain (RBD), which plays a key role in contact with the hACE2 receptor [245]. Fourteen RBD residues bind to the hACE2 receptor. These are Tyr449, Tyr453, Asn487, Tyr489, Gly496, Thr500, Gly502, and Tyr505 are conserved in SARS-CoV-2 RBD, while Leu455, Phe456, Phe486, Gln493, Gln498 and Asn501 are substituted in different forms RBDs of SARS-CoV-1 and SARS-CoV-2 [246]. Biochemical analysis and pseudovirus penetration assay confirmed that the structural features of SARS-CoV-2 RBD have a higher hACE2 binding affinity than SARS-CoV RBDs [247]. The S2 subunit contains regions required for membrane fusion, including two heptad repeats (HR), a transmembrane domain (TM), an internal membrane fusion peptide (FP), and a membrane-proximal external region (MPER) [248].

Infection mechanisms have been investigated and it has been highlighted that the SARS-CoV-2 viral cell-surface spike protein targets hACE2 receptors. Cannalire et al. [249] discussed that the SBP1 peptide has been identified to inhibit the S/ACE2 interaction by targeting the RBD region of the S protein, but the RBD region is more prone to mutation, making it difficult to develop broad-spectrum inhibitors. In this case, the preferred approach is fusion inhibitors targeting the more stable heptad repeats (HRs) involved in the membrane fusion process. Among the peptides described, the EK1C4 lipopeptide exhibits strong S-mediated inhibition of fusion combined with selective broad antiviral activity in cellular assays against SARS-CoV-2 and other relevant CoVs such as MERS-CoV [249]. Another study used a programmatic approach to search and identification based on the hACE2 sequence of ten peptides that have a high potential for interaction with CoV-RBD [250]. Drugs targeting this mechanism are not yet available, but the recent work has demonstrated that peptide hACE2 mimics, creating from the H1 helix and consisting only of natural amino acids, block infection of human lung cells with IC50 in the nM range [251]. It is about eliminating the effect on the renin–angiotensin system and the kinin–kallikrein system in the light of the pathophysiological mechanisms associated with SARS-CoV-2 [252,253]. 

Similarly, to fusion proteins of other coronaviruses, SARS-CoV-2 S is activated by cellular proteases. It is assumed that the S1 and S2 subunits remain non-covalently linked after cleavage [254]. It is known that furin protease of the host cell cleaves the SARS-CoV-2 at the S1/S2 site. Cleavage at the S1/S2 site (residues 669−688) and subsequent cleavage at the S2’ site (residues 808−820) by the TMPRSS2 protease is important for virus penetration into lung cells [255,256]. Tavassoly et al. presented an interesting view that a peptide that is cut between two sites (S1/S2 and S2′) is detached from the virus and enters the intra- or extracellular environment. Subsequently, this peptide can induce some immunological reactions and act as a functional amyloid. The latter hypothesis was supported by the data of the AGGRESCAN program, which identified sites in the S-CoV peptide responsible for the self-aggregation properties [257]. Because the SARS-CoV-2 spike protein is critical for penetration into the host cell, it could be an interesting target protein option for the development of antibacterial peptides. The identification of regions of a protein molecule prone to aggregation/formation of amyloid fibrils is an important step for the development of AMPs acting on the basis of a coaggregation mechanism. For this reason, we used several bioinformatics tools (FoldAmyloid, PASTA 2.0, Waltz and AGGRESCAN), which were previously used [205,207] to predict amyloidogenic sequences in proteins. The results of predicting amyloidogenic regions for the SARS-CoV-2 S protein by various methods are shown in Table 2.

As can be seen from Table 2, amyloidogenic regions identified by any software as amyloidogenic ones are predicted within one amino acid residue, but there are no completely similar predictions for FoldAmyloid, PASTA 2.0, Waltz and AGGRESCAN. However, the prediction of the amyloidogenic region of about 30 amino acid residues at the C-terminus of the SARS-CoV-2 spike protein, common for all four programs, is noteworthy. However, this region corresponds to the transmembrane domain. Interestingly, for the envelope (E) protein in SARS-CoV-1, amyloidogenic sequences at the C-terminus were also previously noted [258]. The same work discusses the possible role of the amyloidogenic sequence in the performance of protein E of its functions of binding to the host cell membranes and formation of ion channels. In our opinion, the amyloidogenic region in the middle of SARS-CoV-2 protein S can be the basis for the development of AMPs acting by the mechanism of directed coaggregation [16]. We hypothesize that anti-CoV peptides can bind to RBD of spike S1 protein (Figure 6). This interaction should reduce the binding affinity of the S1 to the hACE2 receptor of the target cell. A feature of this mechanism is the possibility of viral particles “sticking together”, which can prevent the spread of the virus and reduce the viral load. 

Based on a similar logic, a number of authors tried to synthesize peptide inhibitors of the Spike–hACE2 interaction against SARS-CoV-1 and SARS-CoV-2 viruses [259,260,261,262,263]. As mentioned above, therapeutic peptides have several disadvantages: low bioavailability, short half-life and toxicity. Therefore, lipidation, PEGylation, glycosylation will be useful for improving the pharmacological properties of peptides. In addition, glycosylation can facilitate the specific recognition of viral particle binding sites. For example, how it happens, C-type lectins recognize pathogen patterns [264,265,266].

Recently, promising technologies for creating a vaccine against COVID-19 have been described. In these works, it is proposed to use antigenic peptides developed on the basis of peptide epitopes of viral proteins to inhibit SARS-CoV-2 infection and based on peptide epitopes of T and B lymphocytes to stimulate the human immune response [267]. High-performance in silico technologies currently allows the development of new approaches to the creation of peptide vaccines. Antigenic epitopes found in viral proteins can simultaneously meet several criteria, including immunodominant regions, non-allergenicity, population coverage, and lack of variability (conservatism) for efficient binding and molecular interaction with HLA (human leukocyte antigen) and TLR (toll-like receptor) alleles [238]. To evaluate the effectiveness of peptide vaccines against SARS-CoV-2, the frequencies of the HLA haplotypes can be used to predict the coverage of the developed vaccine. At the same time, peptides are evaluated based on the frequency of HLA haplotypes or HLA alleles in the target population, peptides with undesirable properties are filtered out, which are expected to be glycosylated, identical to peptides in the human proteome, or rapidly degraded [268] Using in silico methods, epitopes of B cells, T cells, and IFN-gamma present on four structural proteins of the virus were mapped, and multi-epitope peptide-based vaccine was developed [269]. It should be taken into account when developing a peptide vaccine that some peptides can provoke an increased production of interleukin 6. In general, the increased mortality of patients with COVID-19 is associated with the induction of the cytokine storm. The work predicted 222 peptide sequences based on the viral spike protein, which can cause increased production of interleukin 6 (IL-6) [270]. The peptide protein kinase inhibitor CK2, previously proven in cancer, can also be used in the treatment of Covid-19. In a small (20 patient) randomized controlled clinical trial, CIGB-325 peptide reduced the mean number of lung lesions. However, in some patients, CIGB-325 may cause itching, redness, and rashes. In the future, it is planned to test the CIGB-325 peptide on more subjects in various combinations with antiviral drugs [271]. Preliminary encouraging results were obtained with the intravenous administration of the CIGB-258 immunomodulatory peptide to critically or severely ill COVID-19 patients. CIGB-258 significantly reduced the levels of biomarkers associated with hyperinflammation, IL-6 and tumor necrosis factor (TNFα) during treatment [272]. Another work suggests the use of annexin A1 Ac2-26 mimetic peptide, which reduces IL-6 production, pain and exudate, which could be a promising treatment in the fight against COVID-19 [273]. In addition, it is assumed that some short peptides, which that have already been developed and tested in the treatment of other diseases, can be used as inhibitors of the SARS-CoV-2 virus, as well as immunomodulators and bronchoprotectors of the pathological process with COVID-19 [274]. In a recent work, a method for preparing a peptide-mRNA vaccine based on engineered peptides synthesized on the binding sequence of the hACE2 receptor, which targets the RBD of the viral S protein, and simultaneously recruits ubiquitin E3 ligases for subsequent intracellular degradation of SARS-CoV-2 in the proteasome, is presented. In a vaccine, mRNA encodes stable perfused forms of the S protein, so that immune cells recognize them [275].

Drug redirection methods for the treatment of SARS-Cov-2 infection are currently being investigated. 300 peptide-like structures from various databases were selected. Using molecular dynamics modeling and docking analysis, the four peptide-like structures demonstrated strong binding affinity for amino acid residues within the SARS-CoV-2 site, the major protease of M^pro^, also called 3CL^pro^ [276,277]. The evolutionary aspect of SARS-CoV-2 should be considered in order to develop peptides that can further target the virus. Thus, in order to select four peptides with strong binding affinity for the main protease SARS-CoV-2, 2765 sequences containing a wide range of mutants from patients with COVID-19 were analyzed by molecular dynamics methods [278]. A computational approach was used to select 15 peptides that showed a higher affinity for RBD of SARS-CoV-2 S protein compared to the α-helix of the hACE2 receptor. It was noted that, in all the detected stable peptide-protein complexes, the Tyr489 and Tyr505 residues in RBD are involved in interaction, which suggests that they are critical for the binding of the discovered antiviral peptides to RBD [279]. After modeling based on the key interacting motifs of the spike protein, four new synthetic SARS-BLOCK™ peptides have been developed and characterized, which can serve as a combined therapeutic, immune and prophylactic agent against SARS-CoV-2. The technology may be interesting if it is possible to show the low cytotoxicity of the developed peptides [280]. 

Since the COVID-19 pandemic was announced, various options have been proposed to combat the infection. Recently, reviews have appeared that draw attention to the prospects for the development and use of antiviral peptides in the therapy of SARS-CoV-2, SARS-CoV-1, MERS-CoV and other respiratory viruses [281]. In particular, there was a point of view that synthetic or natural AMPs can be used to reduce the viral load on cells by blocking the contacts of the virus with receptors on the cell surface [22]. The use of protein/peptide inhalers, due to their prolonged action, higher efficacy, lower systemic availability and minimal toxicity, may be an effective approach for the treatment of SARS-CoV-2 [282]. Modification of existing natural AMPs is an attractive approach, since it can lead to the creation of new antiviral therapeutic agents [11]. The study shows the promise of using nisin, a dietary AMP produced by lactic acid bacteria, which can bind to hACE2, competitively inhibiting RBD SARS-CoV-2 [283]. Against COVID-19, it is proposed to use an AMP such as lactoferrin (LF), which is an iron-binding glycoprotein. However, it should be noted that the antiviral mechanisms of LF differ from virus to viruses. Lactoferrin can enhance the host’s immunity against viral infection or bind directly to the viral particle or receptors and heparan sulfate proteoglycans on the cell-surface of the host cell [284,285]. Interestingly, a recent review suggests the use of peptide BPP-10c (Glu-Asn-Trp-Pro-His-Pro-Gln-Ile-Pro-Pro) derived from venom of *Bothrops jararaca* to counteract the effects of COVID-19 [286]. It has been suggested that AMPs that destroy the envelope glycoprotein can be used to stop or disrupt the life cycle of SARS-CoV-2 [287]. AMPs, which are found in the secretion of mesenchymal stem cells (MSCs), have antimicrobial properties, the ability to attenuate cytokine storms, and, therefore are an attractive approach to prevent the COVID-19 pandemic [288].

The influence of the coronavirus on the course of Alzheimer’s disease has been disclosed. It has been shown that Alzheimer’s disease can be triggered by various bacterial diseases, as well as certain viruses, albeit indirectly. The hypothesis is relevant due to the fact that many patients with COVID-19 have central nervous system disorders and cerebrovascular diseases. There is no data on the penetration of the virus into the human central nervous system, but this is important due to the presence of hACE2 expression in human nerve cells, and the virus can also enter the cerebrospinal fluid through the choroid plexus of the ventricles [289]. Interestingly, Aβ(1-42) peptide associated with Alzheimer’s disease may have antiviral and, in general, antimicrobial activity. AMPs have proven to be antiviral therapy that can be effective against infection and be used to reverse the effects of virus-induced inflammation. However, it is known that viral infections are often accompanied by bacterial infections. COVID-19 is no exception, and some patients die from bacterial co-infection. It can be assumed that the SARS-Cov-2 virus has a mechanism to suppress the production of the host’s own AMPs [290].

AMPs have immunomodulatory effects, with some activating and others suppressing inflammation, which should be considered when selecting candidates. To select specific immunogenic epitopes and accelerate vaccine development, the SARS-CoV-2 genes were analyzed for B and T cell epitope candidates. In addition, the authors propose to increase the low immunogenicity of a peptide vaccine by including an adjuvant in its composition and using an effective delivery system based on chitosan or a copolymer of lactic and glycolic acid (PLGA) [291]. It is suggested that peptides derived from the short-palate, lung and nasal epithelial clone-1 (SPLUNC1), alpha-1-antitrypsin (AAT), dornase alfa (DA) and neutralizing human S230 light chain antibodies can be used as anti-adhesion agents, preventing the SARS-CoV-2 virus from interacting with the human host cell. However, the use of the SPLUNC1 peptide may be associated with 33.3% of allergic reactions, based on in silico data [282]. In the review, the authors discuss bioengineering strategies, chemical modifications, and combined approaches to finding solutions that improve the design, development of AMP, and peptide delivery technologies. It has been noted that clinical use of AMP may be hampered by general properties such as sensitivity to environmental conditions, large size, and poor distribution and excretion [292].

## 5. Conclusions

This review describes various peptides exhibiting antimicrobial properties, their structural and physicochemical features, membranolytic mechanisms of their action and intracellular AMP targets, methods and strategies for the search and creation of new molecules. In this work, we have shown that AMPs have a powerful potential to be used both against the gradually accumulating threat of antibiotic-resistant bacteria and against the suddenly emerging COVID-19 pandemic. Undoubtedly, in the future, the number of antimicrobial peptides used in clinical practice will increase. Improvement of algorithms for prediction and development of new peptides, taking into account the physicochemical properties of already existing AMPs, will contribute here. Additionally, chemical modification and creation of analogues of already well-studied peptides (for example, with membrane binding, amyloidogenic or lectin-like properties) open endless opportunities for the rational design of AMPs with various mechanisms of antimicrobial action. In particular, our proposed new mechanism of antimicrobial action of amyloidogenic peptides, associated with targeted coaggregation with the functional protein of the pathogenic organism, may be of interest for use against a wide range of pathogens including bacteria and viruses. The rapid spread of COVID-19 is the most serious threat to global health this century. It is noteworthy that at present, almost any scientist or just a student can search for the structures of virus inhibitors. So, a cyclic peptide inhibitor of COVID-19 has been described using publicly available software and X-ray crystallographic structures [293]. It is likely that SARS-CoV-2 will occupy a certain niche in the human body and will coexist with us for a long time [294]. Therefore, synthetic or natural AMPs are still an important subject of research [295]. And the search for new antimicrobial drugs remains an urgent problem.

## Figures and Tables

**Figure 1 ijms-21-09552-f001:**
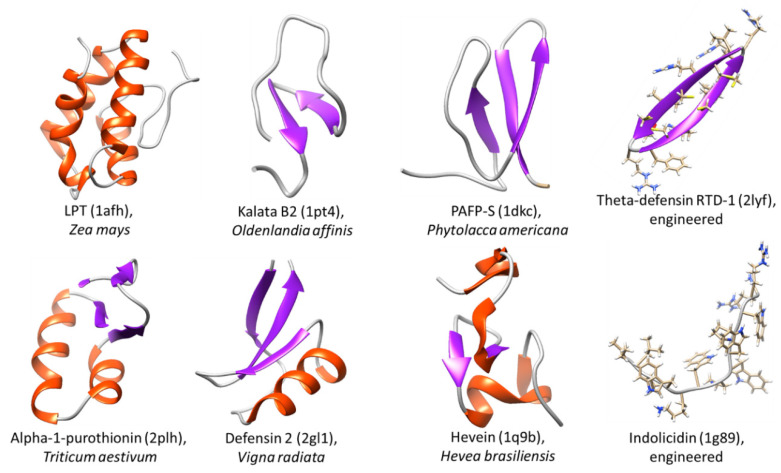
Variety of spatial structures of AMPs.

**Figure 2 ijms-21-09552-f002:**
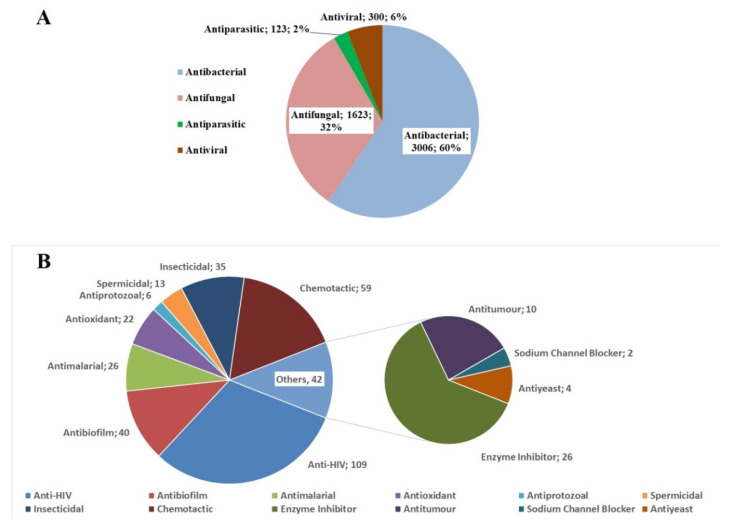
Distribution of AMPs by their biological activity according to dbAMP data: general (**A**) and special (**B**) [79].

**Figure 3 ijms-21-09552-f003:**
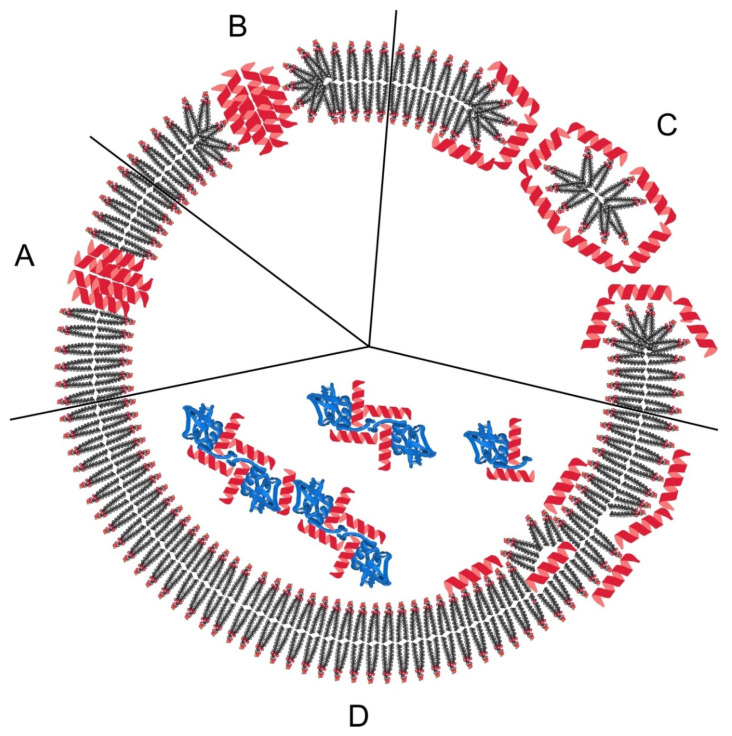
Scheme of diversity of mechanisms for antimicrobial action of AMPs. (**A**) Barrel-stave model; (**B**) Toroidal pore model; (**C**) “Carpet” model; (**D**) Mechanism of direct coaggregation or protein silencing.

**Figure 4 ijms-21-09552-f004:**
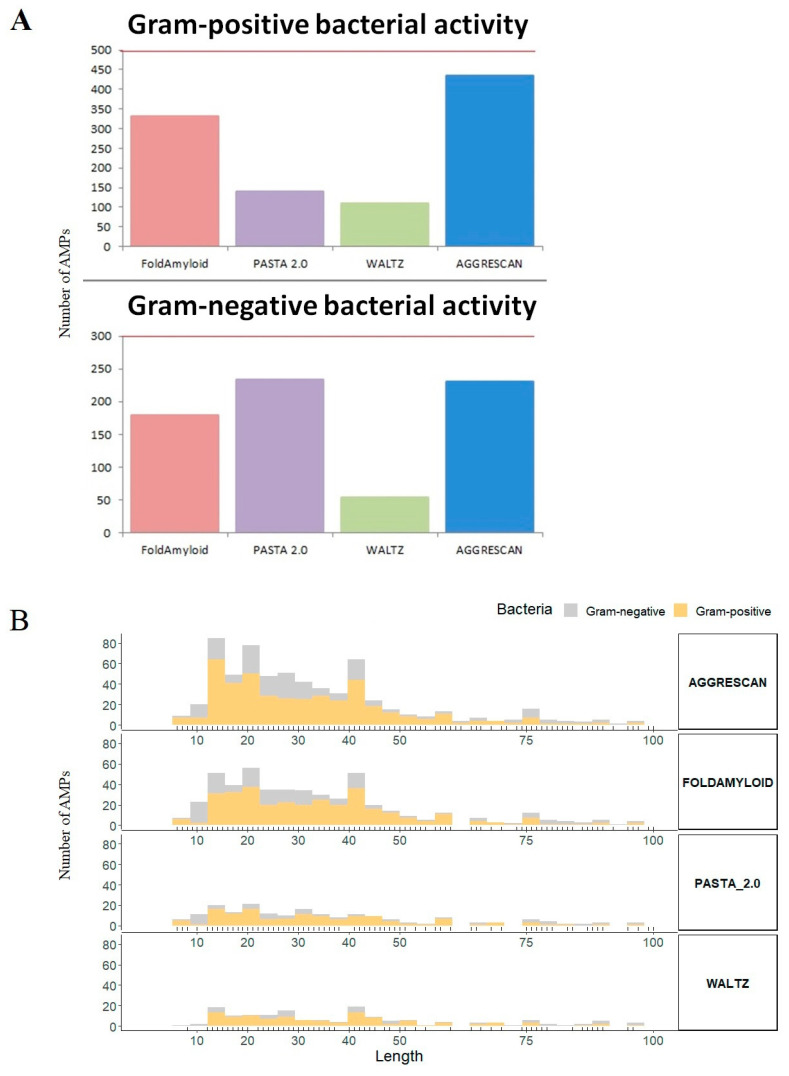
Number of AMPs with predicted amyloidogenic regions by AGGRESCAN [180], FoldAmyloid [181], PASTA 2.0 [182], and Waltz [183]. (**A**) General number (the red line showed count peptides analyzed), (**B**) Length distribution of AMPs.

**Figure 5 ijms-21-09552-f005:**
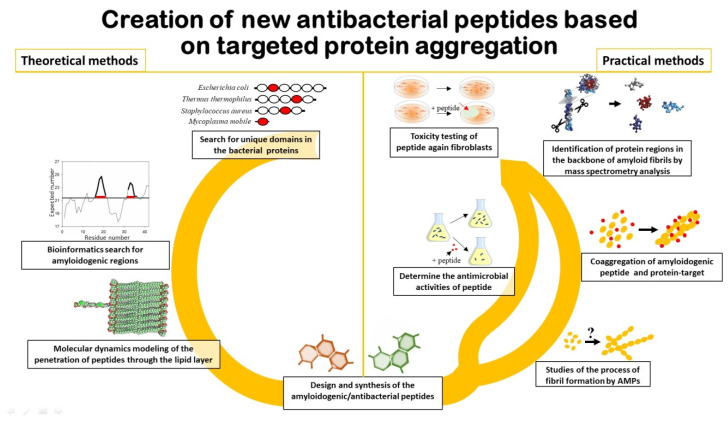
Schematic representation of the main aspects associated with the development of new antibacterial peptides.

**Figure 6 ijms-21-09552-f006:**
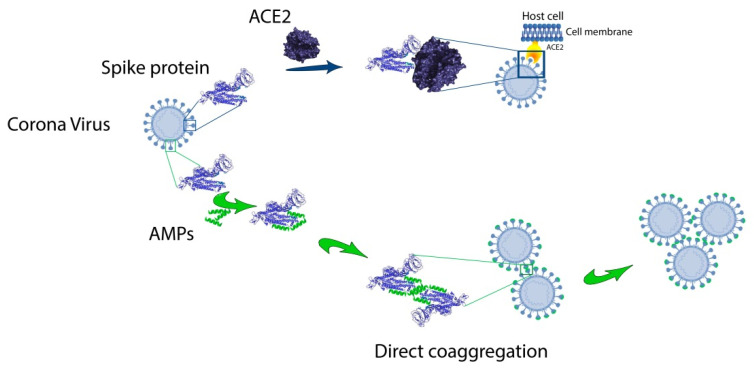
Hypothetical mechanism of direct coaggregation against CoVs.

**Table 1 ijms-21-09552-t001:** Research fields for antimicrobial peptides (AMPs).

Scientific Direction	Description	References
Search and description of new AMPs	This direction is rather diverse and includes from the characterization of new, as yet unknown AMPs using classic biology methods to the creation of AMP search algorithms based on the structural features of peptides and omix data in silico.	[32,38,39,40,41]
Study of physicochemical and structural properties of AMPs	Despite numerous studies of physicochemical properties from 1990 to 2000, much is still unknown about the relationship between physicochemical properties and physiological activity.	[15,42,43,44,45,46]
Generalization of structural properties and prediction of peptides with antimicrobial properties	This area includes the creation of databases of known peptides and algorithms for predicting physiological activity.	[47,48,49,50,51,52,53,54]
Rational design and structure modification	Bacterial resistance to antibiotics creates a need to develop new or modified old molecules to fight bacterial infections.	[3,4,55,56,57]
Study of a narrow biological activity	The specific biological activity of AMP is an important property that requires close attention.	[36,37]
Methodical developments	The results of AMP research over the decades have led to the revisions and modifications of experimental procedures.	[58,59]
Solving application problems in practice.	Improving the effectiveness of AMP can be realized through targeted delivery using non-materials or a combination of dosage forms. Non-drug forms of AMP are proposed to be used as antiseptics to combat dangerous pathogens. Another possible application of AMPs is in biomarkers of parasitic infections.	[33,34,35,60,61,62]

**Table 2 ijms-21-09552-t002:** Amyloidogenic regions predicted by different programs for SARS-CoV-2 spike protein.

Software	Amyloidogenic Regions for SARS-CoV-2 Spike Protein (https://www.uniprot.org/uniprot/P0DTC2)
AGGRESCAN (UAB, Barcelona, Spain http://bioinf.uab.es/aggrescan/)	1–8, 10–16, 31–38, 40–47, 50–56, 58–70, 86–91, 100–109, 114–133, 140–145, 162–167, 189–208, 231–247, 260–271, 303–311, 331–336, 338–354, 362–383, 390–400, 428–437, 448–457, 484–490, 508–520, 584–588, 590–599, 610–617, 666–671, 688–698, 716–727, 729–745, 751–759, 761–768, 783–787, 797–805, 817–833, 853–861, 864–870, 872–914, 961–982, 1000–1013, 1044–1052, 1058–1068, 1097–1101, 1124–1137, 1171–1179, 1207–1249.
PASTA 2.0 (University of Padova, Padova, Italy http://protein.bio.unipd.it/pasta2/)	1214–1244.
Waltz (Switch Laboratory, Leuven, Belgium http://waltz.switchlab.org/index.cgi)	1–9, 88–95, 115–137, 141–146, 199–204, 261–271, 275–280, 365–378, 447–455, 485–496, 537–544, 608–618, 689–697, 703–727, 798–806, 913–940, 969–974, 1003–1015, 1063–1068, 1117–1122, 1175–1179, 1206–1233.
FoldAmyloid (Institute of Protein Research of the Russian Academy of Sciences, Pushchino, Russia http://bioinfo.protres.ru/fold-amyloid/)	1–10, 34–38, 53–69,100–108,116–120, 126–135, 141–146, 174–178, 190–195, 235–244, 264–269, 274–278, 327–331, 348–353, 366–370, 389–395, 431– 436, 450–457, 487–492, 507–519, 539–543, 560–564, 582–586, 609–613, 691–697, 718–722, 738–743, 751– 755, 780–784, 799–804, 819–825, 875–880, 894–907, 993–999, 1005–1014, 1047–1051, 1060–1067, 1101–1105, 1127–1132, 1210–1239.

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
