# Peer review of "Antimicrobial and Amyloidogenic Activity of Peptides. Can Antimicrobial Peptides Be Used against SARS-CoV-2?"

_ijms, 2020, doi:10.3390/ijms21249552_

Round 1
Reviewer 1 Report
I have read the manuscript entitled "Antimicrobial and Amyloidogenic Activity of Peptides. Can Antimicrobial Peptides Be Used Against SARS-CoV-2?" with great interest and I think it is in principle suited for a publication in the International Journal of Molecular Sciences, section: Molecular Microbiology. Unfortunately, I have to answer the question of the authors in the title at the moment with: “I really do not know.” It is documented in the literature that glycobiology plays a prominent role in this context (e. g. when discussing heveins and defensins with their lectin-like functions). The authors are leaving out this aspect completely, except mentioning the word “glycoprotein”. May be their review article becomes a valuable scientific contribution after a major revision considering this aspect.
Author Response
Point 1: I have read the manuscript entitled "Antimicrobial and Amyloidogenic Activity of Peptides. Can Antimicrobial Peptides Be Used Against SARS-CoV-2?" with great interest and I think it is in principle suited for a publication in the International Journal of Molecular Sciences, section: Molecular Microbiology. Unfortunately, I have to answer the question of the authors in the title at the moment with: “I really do not know.” It is documented in the literature that glycobiology plays a prominent role in this context (e. g. when discussing heveins and defensins with their lectin-like functions). The authors are leaving out this aspect completely, except mentioning the word “glycoprotein”. May be their review article becomes a valuable scientific contribution after a major revision considering this aspect.
Response 1: We would like to thank the reviewer for your efforts and constructive suggestions and comments. We revised our review and, among other things, added more details in the text and references to the works that consider the aspect of glycobiology and lectin-like functions of AMPs in the fight against pathogens (lines 71-74, 127-135, 770-773). We answer yes to the question "Can Antimicrobial peptides be used Against SARS-CoV-2?" We have added a new “Introduction” section, a new Table 1, a new paragraph in the conclusion section, and new Figure 6 for better understanding.
Reviewer 2 Report
Kurpe et al have tried to give a characterization of AMPs which is already known in the literature. The review does not evoke any interest as there is a significant lack of recent work on antimicrobial peptides. They have written only the things which are already known in the field.
After reading through 16 pages of this review, one reaches the topic of SARS-CoV-2. Although the investigators have done some work to identify amyloidogenic regions and prediction for SARS-CoV-2 spike protein, it's not enough to answer the question about the utility of AMPs against this deadly virus. Even the authors seem to be hesitant to answer this question till the end of their manuscript.They need to add more convincing results to see if AMPs can be used against SARS-CoV-2.
Author Response
Point 1: Kurpe et al have tried to give a characterization of AMPs which is already known in the literature. The review does not evoke any interest as there is a significant lack of recent work on antimicrobial peptides. They have written only the things which are already known in the field.
Response 1: We would like to thank the reviewer for reading our work. In order to increase the interest of readers of the International Journal of Molecular Sciences, we have added several new studies on antimicrobial peptides to the review. Of the nearly three hundred cited works, more than a hundred interesting, in our opinion, studies have been published this year alone.
Point 2: After reading through 16 pages of this review, one reaches the topic of SARS-CoV-2. Although the investigators have done some work to identify amyloidogenic regions and prediction for SARS-CoV-2 spike protein, it's not enough to answer the question about the utility of AMPs against this deadly virus. Even the authors seem to be hesitant to answer this question till the end of their manuscript. They need to add more convincing results to see if AMPs can be used against SARS-CoV-2.
Response 2: Thank you very much for your valuable comment. We significantly revised the structure of the manuscript and added an Introduction section (lines 34-82) to explain the relevance and usefulness of antimicrobial peptides against SARS-CoV-2. In addition to considering published AMP studies, we also described our version of using the amyloidogenic properties of peptides to develop new AMPs. Our proposed mechanism for direct coaggregation or protein silencing can be useful for combating a wide range of pathogens, including SARS-CoV-2 (lines 755-766 and Figure 6). In the Conclusion section, based on the revised version of the review, we can confidently state that antimicrobial peptides have great potential and can be used against SARS-CoV-2.
Round 2
Reviewer 1 Report
The authors have significantly improved their review article regarding the field of glycobiology. However, the important aspects of sialic acids is still not considered:
R. Zhang, T. Eckert, T. Lütteke, S. Hanstein, A. Scheidig, A. M. J. J. Bonvin, N. E. Nifantiev, T. Kožár, R. Schauer, M. A. Enani, H.-C. Siebert (2015) Structure-function relationship of antimicrobial peptides and proteins in respect to contact-molecules on pathogen surfaces. Curr. Top. Med. Chem., 16, 89-98.
A minor revision is therefore necessary.
Reviewer 2 Report
This is a much-improved version than the previous one. The authors have thoroughly rewritten the manuscript to make it more interesting. Congratulations on that. I recommend accepting the manuscript in its current form.